# Evaluation of the Hepatotoxicity of the Zhi-Zi-Hou-Po Decoction by Combining UPLC-Q-Exactive-MS-Based Metabolomics and HPLC-MS/MS-Based Geniposide Tissue Distribution

**DOI:** 10.3390/molecules24030511

**Published:** 2019-01-31

**Authors:** Yunji Wang, Fang Feng

**Affiliations:** 1Key Laboratory of Drug Quality Control and Pharmacovigilance, Ministry of Education, China Pharmaceutical University, Nanjing 210009, China; wangyunji1984@126.com; 2Department of Pharmaceutical Analysis, China Pharmaceutical University, Nanjing 210009, China

**Keywords:** Zhi-Zi-Hou-Po decoction, hepatotoxicity, metabonomics, geniposide, tissue distribution

## Abstract

With traditional Chinese medicine (TCM) becoming widespread globally, its safety has increasingly become a concern, especially its hepatoxicity. For example, *Gardenia jasminoides* Ellis is a key ingredient in the Zhi-Zi-Hou-Po decoction (ZZHPD), which is a commonly-used clinically combined prescription of TCM that may induce hepatoxicity. However, the underlying toxicity mechanism of ZZHPD is not fully understood. In this study, a plasma metabolomics strategy was used to investigate the mechanism of ZZHPD-induced hepatotoxicity through profiling entire endogenous metabolites. Twenty-four Sprague-Dawley rats were randomly assigned into four groups, which were orally administered with 0.9% saline, as well as 2.7 g/kg/day, 8.1 g/kg/day, or 27 g/kg/day of ZZHPD for 30 consecutive days, respectively. Biochemical assay and metabolomics assay were used to detect serum and plasma samples, whilst histopathological assay was used for detecting liver tissues, and the geniposide distribution in tissues was simultaneously measured. The results showed that the concentration of 20 metabolites linked to amino acid, lipid, and bile acid metabolism had significant changes in the ZZHPD-treated rats. Moreover, toxic effects were aggravated with serum biochemical and histopathological examines in liver tissues as the dosage increased, which may be associated with the accumulation of geniposide in the liver as the dosage increased. Notably, our findings also demonstrated that the combined metabolomics strategy with tissue distribution had significant potential for elucidating the mechanistic complexity of the toxicity of TCM.

## 1. Introduction

The Zhi-Zi-Hou-Po decoction (ZZHPD) is a typical traditional Chinese medicine (TCM) prescription that was originally described in Shang-Han-Lun, and include *Gardenia jasminoides* Ellis (Zhi-Zi), *Citrus aurantium* L. (Zhi-Shi), and *Magnolia officinalis* cortex (Hou-Po). In clinical practice, the ZZHPD has been widely used for the treatment of psychiatric illnesses, especially major depressive disorder (MDD) [1,2], a chronic disease. Zhi-Zi, the main ingredient of the ZZHPD, has been reported to have significant antidepressant effects [3,4]. Clinically, the dosage of Zhi-Zi is often increased to achieve the curative effect more rapidly. However, concerns have been raised about the safety of herbal prescriptions [5], where an increasing number of reports suggest that Zhi-Zi induces liver toxicity [6,7]. However, there is no scientific evidence of the toxicity behind the Zhi-Shi [8,9] and Hou-Po [10] extracts. In China, Zhi-Zi is also used as a food ingredient or a dietary supplement [11], which is mixed in with tea or healthcare foods. The daily dosage of six to 10 g was set by the Chinese pharmacopoeia [12]. A daily dosage of 30 g or more may cause liver damage [13]. Geniposide is a key bioactive component of Zhi-Zi. It not only reduces amyloid β peptide-induced toxicity and promotes neurite outgrowth [14], indicating its antidepressant-like effects [15], but it can also cause liver toxicity from overdosing [16,17,18,19]. Prior research has not investigated the concentration changes of geniposide in different rat tissue or explored its correlation with ZZHPD-induced hepatotoxicity. Unfortunately, neither the ZZHPD-induced hepatotoxicity nor the methods for early detection, such as the hepatotoxicity mechanism, have been investigated. Therefore, there is an urgent need to examine the relationship between geniposide and ZZHPD-induced liver damage, and identify the toxic biomarkers for early monitoring the occurrence of the liver injury.

In addition, metabonomics has focused on a holistic investigation of endogenous small molecule metabolites, which enable living systems to respond to external stimuli, in the study of drug safety evaluation, toxicity prediction, and disease diagnosis, by comprehensively tracking the changes of the small molecule metabolites [20,21,22]. Recently, a number of metabonomic studies were carried out for the prediction of TCM-induced hepatotoxicity, such as ¹H-NMR metabolomics application in the acute toxicity evaluation of *Clinacanthus nutans* water leaf extract [23]. LC-MS and GC-MS platforms based on serum metabolomics were applied to identify potential endogenous metabolites for hepatotoxicity induced by *Polygoni Multiflori Radix* [24], whilst UPLC-QTOF-MS was applied in a study to discover the characteristics of potential biomarkers in rats treated with *Xanthii Fructus* using urinary metabonomics [25]. However, there has been no report of metabonomic studies of ZZHPD hepatoxicity in plasma.

In the present study, plasma metabonomic based on UPLC-MS was applied to investigate the metabolic profile and hepatotoxicity mechanism of three doses of the ZZHPD, over a 30-day intragastric administration. The potential biomarkers were identified through a pathway analysis of ZZHPD-induced liver damage. Meanwhile, a sensitive and selective HPLC-MS/MS method was successfully established and verified, according to the United States Food and Drug Administration (FDA) regulations on bioanalytical methods for the quantitative determination of tissue distribution of geniposide in the overdose of ZZHPD intragastric administration in rats. These findings may be helpful for further investigations of the hepatotoxicity mechanism of ZZHPD, as well as provide a reference basis for the rational clinical use of the drugs.

## 2. Results

### 2.1. Toxicity Effects and Biochemical Analysis

The chronic toxicity of ZZHPD could be intuitively deduced from symptoms, such as hair pollution, hair loss and thinning, and loose stools in the 27 g/kg/day ZZHPD group. Compared to the control group, the activities of the rats were obviously decreased after oral administration in the 27 g/kg/day ZZHPD group. This effect was not observed in the 2.7 g/kg/day and 8.1 g/kg/day ZZHPD groups. The rats were dissected at the end of the administration. The kidney colors of the rats in the 8.1 g/kg/day and the 27 g/kg/day ZZHPD group were dark green, while the liver color of the rats in the 27 g/kg/day ZZHPD group was abnormally dark brown. However, these symptoms were not observed in the 2.7 g/kg/day ZZHPD group. There was no significant difference in the food intake, water consumption, and body weight in the 2.7 g/kg/day and 8.1 g/kg/day groups, but there were significant differences in the 27 g/kg/day ZZHPD group (Figure 1).

To investigate the ZZHPD toxicity, the biochemical parameters in serum from the ZZHPD orally-administrated rats were measured at the end of the experiments (Table 1). Compared with the control group, no significant changes in the parameters were observed in the ZZHPD-treated rats from the 2.7 g/kg/day ZZHPD group. Meanwhile, the levels of alkaline phosphatase (ALP), aspartate aminotransferase (AST), alanine aminotransferase (ALT), total bilirubin (TBIL), serum creatinine (CREA), urea nitrogen (BUN), total protein (TP), and total cholesterol (TC) in the ZZHPD-treated rats from the 8.1 g/kg/day ZZHPD group had changed significantly. Indeed, it is well-known that ALP, AST, and ALT are the main indicators of liver function. The TBIL is also used to diagnose liver disease or biliary tract abnormalities. The levels of creatinine and BUN in serum represent the kidney function. The serum total protein is an important item in clinical biochemical detection, and it has many functions, such as maintaining the normal blood colloid osmotic pressure and PH, transporting various metabolites, regulating the physiological functions of the transported substances, and relieving the substances’ toxicity, immunity, and nutrition. The serum level of TC is a routine project in lipid analysis, and it is closely related to cardiovascular and cerebrovascular diseases such as atherosclerosis, coronary heart disease, and stroke. There were significant changes in the above parameters of the serum in the 8.1 g/kg/day and 27 g/kg/day ZZHPD groups, out of the four groups. These results indicated rat liver function damage induced by the ZZHPD oral administration in the 8.1 g/kg/day and 27 g/kg/day ZZHPD groups over 30 consecutive days.

### 2.2. Histopathological Assay

We found that, of the pivotal indicators of liver function, the ALP, AST, and ALT had higher levels in the animals treated by the ZZHPD than in the control animals. Thus, the histomorphology of the liver sections was further investigated using hematoxylin–eosin staining. Compared to the control group, the hepatic tissue sections of the rats treated using the ZZHPD in the 2.7 g/kg/day ZZHPD group showed normal integrated hepatic cells and lobular architecture with hepatic central veins (Figure 2a,b). Meanwhile, a slight degree of degeneration of the hepatocyte vacuolation and the infiltration of inflammatory cells into the portal area were observed in the 8.1 g/kg/day ZZHPD group (Figure 2c). Dose-dependent histopathological changes were distinctly observed in the liver tissues of the high-dose group of the ZZHPD, including the proliferation of intrahepatic bile duct, the necrocytosis of hepatocytes, and a few hyaline droplets in the liver (Figure 2d). Therefore, the results indicated that liver injury might occur during oral administration in the ZZHPD-treated animals from the 8.1 g/kg/day and 27 g/kg/day ZZHPD groups for 30 consecutive days, and it displayed a positively dose-dependent manner.

### 2.3. Tissue Distribution Study of Geniposide

#### 2.3.1. Method Validation

The HPLC-MS/MS method was validated for selectivity, linearity, sensitivity, precision, accuracy, matrix effect, extraction recovery, dilution, carryover, and stability following the guidelines of the FDA (United States Food and Drug Administration). Geniposide and bezafibrate (internal standards, IS) appeared on the chromatograph at 1.19 min and 2.34 min, respectively, with no interfering peaks (Figure 3). The standard curve linear range of the geniposide in plasma and tissue was 5.0 to 5000 ng/mL, with a weighted (1/X2) least squares linear regression and linear regression coefficient that was greater than 0.99 in each run. The precision and accuracy of the quality control plasma and tissue samples (5.0 ng/mL, 12.5 ng/mL, 400 ng/mL, and 4000 ng/mL, respectively, *n* = 6) of geniposide were determined over three continuous validation batches. The precision and accuracy of the intraday and interday geniposide in plasma and tissue demonstrated that the values met the requirements for validation (Appendix A). The matrix effect was assessed using the ratio of the peak area of blank tissue homogenate and plasma spiked after extraction to standard solutions. The extraction recovery of geniposide was determined by comparing the peak ratio of analyte-spiked blank tissue homogenate and plasma before extraction to be spiked into the solution extracted from the blank tissue homogenate and plasma. The result demonstrated that there were no substances affecting the ionization of geniposide or that had high extraction efficiency. A stability study was investigated under different conditions as follows: incubation at room temperature (25 °C) for 12 h and 15 h, three freeze-thaw cycles, after sample preparation at 8 °C for 53 h, and storage at −80 °C for 90 days. The results of the stability study indicated that geniposide was stable under the above experimental conditions (Appendix A).

#### 2.3.2. Tissue and Plasma Sample Analysis of Geniposide

To investigate the substance foundation of ZZHPD-induced hepatotoxicity, and also the relationship between toxicity and dosage with the variation of geniposide in the rat tissues, geniposide was determined using the developed method. As shown in Figure 4, after oral administration, geniposide was distributed to the plasma and various tissues, including the heart, liver, spleen, lung, kidney, and brain. The results indicated that geniposide was distributed rapidly and widely in these tissues in the order of kidney > plasma > liver > lung > spleen ≈ heart > brain across the three dosages. The high dosage group had a higher concentration of geniposide in the kidney and liver at one hour, but this showed no significant alteration at four hours.

### 2.4. Validation of the UPLC-Q-Exactive Orbitrap-MS

In the current study, metabolomics studies were performed using the UPLC-Q-Exactive Orbitrap-MS of plasma in both the positive and negative mode. To evaluate the system’s stability, six species of ions were selected for monitoring as extracted ion chromatograms, throughout the entire analysis batch of the quality control (QC) samples in both the positive and negative ion modes. To cover the whole analysis batch, we extracted the ion chromatographic peaks of the six ions with high abundance ratios, different retention times, and spectral regions. The exact mass and retention time pairs of these ions in the plasma samples were *m*/*z* 130.0860/1.28 min, *m*/*z* 232.9960/6.49 min, *m*/*z* 348.2239/8.90 min, *m*/*z* 631.3188/14.28 min, *m*/*z* 510.3537/14.48 min, and *m*/*z* 482.3225/15.45 min in the positive ion mode; and *m*/*z* 253.0297/4.89 min, *m*/*z* 113.0229/5.27 min, *m*/*z* 305.2386/15.11min, *m*/*z* 496.2726/9.08 min, and *m*/*z* 794.9487/17.19 min in the negative ion mode, respectively. The relative standard deviations (RSDs) of the retention times and peak intensity of the six selected ions were between 1.79–8.51% and 0.64–12.06%, respectively. The injection precision was evaluated by analyzing six successive injections of the same QC sample. For the plasma samples, the RSDs ranged from 0.79% to 3.26% for the peak intensity and from 0.02% to 1.21% for the retention time in the positive ion mode, and from 0.24% to 4.18% for the peak intensity and from 0.04% to 1.53% for the retention time in the negative ion mode.

### 2.5. Metabolic Profiling Analysis

The plasma metabolic fingerprints were characterized using UPLC-MS. The typical total ion chromatograms (TIC) of the rat plasma obtained in the positive and negative mode, respectively, are shown in Figure 5. Although some variations in the metabolites could be observed from the chromatograms of the four groups, a detailed result containing t_R_-*m*/*z*, ion intensity, and sample code, exported by the XCMS Online platform, was further investigated by means of the unsupervised principal component analysis (PCA) to examine the global metabolism. Figure 6 illustrates the score plot. In the score plot, the four groups were clearly separated, which indicated that the endogenous metabolites were significantly changed in the three treatment groups and in the control group, where the 2.7 g/kg/day ZZHPD group was much closer to the control group. The potential biomarkers were extracted from the S-plot (Figure 7a,b) and obtained through a comparison between the control group and the 27 g/kg/day ZZHPD group, based on partial least squares-discriminate analysis (PLS-DA). The 27 g/kg/day ZZHPD group was different from the other treatment groups and the control group, which suggested 27 g/kg/day ZZHPD toxic effects on the rats. Compared to the unsupervised PCA, the supervised orthogonal partial least squares discriminant analysis (OPLS-DA) was able to clearly differentiate the specific variables amongst these groups. In the scatter plot of the OPLS-DA (Figure 8), we found that samples from the 27 g/kg/day ZZHPD group and the control group were distributed in different areas, which indicated a differentiation in metabolic patterns between the control group and the 27 g/kg/day ZZHPD group.

### 2.6. Biomarker Identification

The identification of potential biomarkers in the discrimination of the treatment group from the control group could provide an important advancement and enhance the understanding of the mechanism of the ZZHPD’s toxicity. All of the ions detected by the UPLC-ESI-Q-Exactive Orbitrap-MS were sorted in a decreasing order according to their variable importance in the projection (VIP) and p values. We selected ions based on the VIP value derived from the OPLS-DA model, which showed a noticeable change in the control group compared to the treatment groups as potential biomarkers. The identification of potential biomarkers relied on the literature, possible chemical constitutions, retention time, precise mass, and MS/MS data, which were screened to determine the potential structures of the ions in the Human Metabolome Database (http://www.hmdb.ca/), Metlin database (https://metlin.scripps.edu/), and MassBank (http://www.massbank.jp/), which were developed by metabolomics. The identified metabolites were further analyzed using t-tests to evaluate the statistical significance between the three dose groups and the control group (*p* < 0.05). Twenty compounds were identified amongst the four groups as the hepatoxicity of the ZZHPD biomarkers, which are listed in Table 2.

### 2.7. Metabolomic Pathway Analysis

To discover the significantly relevant hepatic toxicity metabolic pathways, we performed a metabolomic pathway analysis on the 20 potential biomarkers using MetaboAnalyst 4.0 (http://www.metaboanalyst.ca), which integrates enrichment analysis and pathway topology analysis. Figure 9 shows that phenylalanine, tyrosine, and tryptophan biosynthesis were the most relevant pathways affected by ZZHPD, with the impact value of 0.5. In addition, comparing the ZZHPD treatment groups with the control group, several important pathways were identified: arachidonic acid metabolism, histidine metabolism, sphingolipid metabolism, cysteine and methionine metabolism, and tyrosine metabolism. Moreover, we performed a metabolic network using the Kyoto Encyclopedia of Genes and Genomes (KEGG, http://www.kegg.jp) pathway database. Our data showed that the comprehensive metabolic changes in rats from of ZZHPD-induced hepatotoxicity were mainly related to amino acids, bile acids, lipids, and energy metabolism.

## 3. Discussion

Identifying critical metabolites, as well as the correlation pathways that are disturbed by drug induced toxicity in the body, through using modern technological means was helpful to our understanding of the mechanisms associated with drug-induced liver damage. In the present study, LC/MS-based metabonomic methods coupled with biochemical analysis and a histopathology method were used to examine the effects induced by the ZZHPD. Meanwhile, to explain the basic substances of the drug’s toxicity, the LC-MS/MS method was applied to investigate the tissue distribution of geniposide.

With the long-term administration of the ZZHPD, liver injury was observed in the 8.1 g/kg/day and 27 g/kg/day ZZHPD groups with the rise of serum biochemical indices, such as AST, ALT, ALB, and ALP, as confirmed by the hepatic cell of inflammatory infiltration and necrosis in the liver histopathology examination. TBIL has been used to detect cholestasis, and it includes indirect/unconjugated bilirubin (IBIL) and bilirubin (DBIL). TBIL has been used to detect IBIL, where it is generated by the degradation of heme in aged erythrocytes; then, it is circulated to the liver to be converted into DBIL. DBIL is excreted through the canalicular membrane into the bile duct, and then directly secreted into the duodenum [26]. TBIL levels were elevated in the 8.1 g/kg/day and 27 g/kg/day ZZHPD groups, which we considered to be due to the failure to excrete bile acids because of the hepatic function deficits. The levels of BUN in serum represent the kidney function. In the 8.1 g/kg/day and 27 g/kg/day ZZHPD groups, it was shown that kidney function had also been affected. Compared to the control group, although there were significant differences in the biochemical indexes between the 27 g/kg/day ZZHPD group and the control group, there were no significant toxic reaction symptoms found in the rats under daily observation. Therefore, liver injury occurred in the 8.1 g/kg/day ZZHPD group, whose observation was not easy to find. To elucidate the chronic hepatotoxicity of ZZHPD, the changes in toxicity and the effecting substance geniposide of the ZZHPD should be monitored during the occurrence of liver injury. Hence, we conducted a tissue distribution study of geniposide after the ZZHPD administration for 30 days.

In our current study of tissue distribution, we observed that after oral administration of the ZZHPD at a dosage of 27 g/kg/day, geniposide had accumulated in the liver, which may be the basis for the production of liver toxicity. The kidney is the main discharge site of such polar metabolites, and where the concentration of geniposide is the highest. This result was in agreement with the findings of previous studies on the tissue distribution of geniposide following intravenous and peroral administration to rats [27]. However, we did not observe tissue lesions in the tissue section of the kidney in the 8.1 g/kg/day and 27 g/kg/day ZZHPD groups; rather, a dark green appearance was found in the anatomical process. We could speculate that hepatotoxicity may not only be caused by the accumulation of geniposide, but it also may be related to its biotransformation [28]. Being a hydrophilic glycoside, geniposide cannot be through the intestine membrane, and it seems indispensable to be hydrolyzed into its aglycone genipin by β-d-glucosidases in the intestines before absorption [29]. Due to the presence of an acidic hydrogen (p*K*_a_ 6.3), genipin was further metabolized by the intestine and liver through a sulfation and glucuronidation reaction before circulation. Given this, the concentrations of geniposide in the plasma and liver were not high, which accounted for the presystemic metabolism in the gut lumen, and the first pass effect of geniposide in enterocytes and hepatocytes [30]. A previous study reported that the concentrations of genipin in rat liver tissue were much higher than in the kidney and plasma after the oral administration of geniposide, which suggested that genipin may contribute more to liver injury [31]. Moreover, CYP450 isoezymes, which mainly exist in the liver and extrahepatic tissues, play an important role in the metabolism of about 90% of drug oxidation and reduction [32]. Genipin has significant inhibition on the CYP3A4 and CYP2C19 expression of mRNA and protein [33]. Flavonoid glycoside components in the Zhi-Shi, such as narirutin, hesperidin, naringin, and neohesperidin, also have relatively strong inhibitory effects on CYP3A4 [34,35], which can alter the pharmacokinetics of geniposide and genipin, leading to an increase in systemic exposure. The 8.1 g/kg/day and 27 g/kg/day ZZHPD groups were more likely to slow down the metabolism of geniposide by inhibiting the activity of the CYP450 enzyme, such that the geniposide and genipin accumulated in the liver, leading to a toxicity reaction.

Sphinganine, C16-sphinganine, LysoPC (20:4), LysoPC (18:1), LysoPC (16:0), LysoPE (0:0/20:4), and arachidonic acid were involved in the lipid metabolism pathways (Figure 10). Phosphoglyceride and sphingolipids were the main components of the lipid metabolic pathway in organisms. Phosphoglyceride was divided into several categories, such as the phosphatidylcholine (PC), phosphatidylethanolamine (PE), etc. Phospholipase A2 (PLA2), which is a rate-limiting enzyme of arachidonic acid (AA), can hydrolyze PC to produce AA and LysoPCs. Sphinganine and C16-sphinganine are sphingolipids in sphingolipids biosynthesis and metabolism, which are lipid-signaling molecules that are related to inflammation processes. The sphingosine is produced from ceramide by ceramidases. Through sphingosine kinases, it can be further phosphorylated to sphingosine-1-phosphate (S1P) [36]. Additionally, S1P participated in the gene regulation of adhesion molecules in inflammation progression and tissue damage [37]. S1P significantly induced cyclooxygenase-2 (COX-2) expression and prostaglandin E2 (PGE2) production in human granulosa cells through an S1P1/3-mediated yes kinase-associated protein (YAP) signaling pathway [38]. COX-2 is the main speed limit for PGE2 synthesis enzymes, which are mainly produced under the action of proinflammatory factors, such as cytokines and growth factors. Therefore, the up-regulation of sphinganine will increase the production of S1P and further induce the expression of COX-2 and PEG2, which cause an inflammatory response. In the current study, compared to the control group, during the long-term administration of ZZHPD, sphinganine and C16-sphinganine were significantly increased in the 8.1 g/kg/day and 27 g/kg/day ZZHPD groups, which could be a factor of the inflammatory response. At the phosphoglyceride pathway metabolic, lysoPCs, lysoPE, and AA significantly changed in the 27 g/kg/day ZZHPD group compared to the control group. The results showed that the AA significantly increased in the 27 g/kg/day ZZHPD group compared to the control group. We speculate that it was probably related to the PLA2 activation. Endogenous lysoPCs have important functions in maintaining homeostasis in organisms, that is, the product of low-density lipoprotein (LDL) oxidation can be oxidized by a series of factors, including free radicals [39]. LysoPE (0:0/20:4), the degradation product of PE, has an important function in membrane structure and energy reserves. After the massive destruction of membranes, PE was increased, and it was indicated by cell death, which was observed during the histological examination of the livers [40]. The lysoPCs significantly increased in the 8.1 g/kg/day and 27 g/kg/day ZZHPD groups in the rat plasma. The explanations for the above results include: (1) the 8.1 g/kg/day and 27 g/kg/day ZZHPD groups can induce the activity of PLA2 and result in the increase of lysoPCs in rat plasma; (2) the 8.1 g/kg/day and 27 g/kg/day ZZHPD groups showed free radical activity and damaged LDL from peroxidation by free radicals; (3) the 8.1 g/kg/day and 27 g/kg/day ZZHPD groups can inhibit the activity of paraoxonase, which protects LDL from oxidation; and (4) the LysoPE increase in the plasma could be the massive liver cell death induced by the ZZHPD, and LysoPEs are enriched in the inner leaflet of the mitochondria. The dosage of 27 g/kg/day ZZHPD caused severe damage to the mitochondria.

The disturbance of plasma, including the metabolism of amino acids, was observed in rats in the 8.1 g/kg/day and 27 g/kg/day ZZHPD groups, which could have resulted from the hepatic induced by the ZZHPD treatment. Homocysteine is a sulfuric amino acid, which can generate glutathione through the transsulfuration pathway. Glutathione is the major intracellular redox buffer in the liver, and it is critical for the hepatic detoxification of xenobiotics and other environmental toxins [41]. The up-regulation of homocysteine may reflect damage to the liver, and the body produces more homocysteine to reduce detoxification. Compared to the control group, there was an increase of hippuric acid in the 8.1 g/kg/day and 27 g/kg/day ZZHPD groups. Hippuric acid is considered as a potential endogenous antioxidant in organisms, which is produced by combining glycine with benzoic acid during cytochrome P450 catalysis [42]. The content of hippuric acid changed in the 8.1 g/kg/day and 27 g/kg/day ZZHPD groups, which was closely related to liver injury. Vinylacetylglycine is an important metabolite of fatty acid. The decreased vinylacetylglycine may be because quercetin can alleviate the lipid metabolic disorder [43]. In the 8.1 g/kg/day and 27 g/kg/day ZZHPD groups, vinylacetylglycine was an up-regulation, and it showed that lipid metabolic disorder had occurred.

Bile acid is the general term of choleic acid in bile, which can directly reflect the metabolic state of the liver. However, abnormal bile acid metabolites can regulate the intestinal liver circulation and liver disease progression of bile acid by affecting the functions of immune cells, liver cells, and intestinal flora [44]. Cholic acid belongs to bile acid, and the up-regulation in the 27 g/kg/day ZZHPD group showed that the liver cells were damaged, and that bile acid metabolism was affected (Figure 9).

In this work, we outlined a metabolomics strategy combined with serum biochemical analysis, histopathological examination, and tissue distribution of geniposide to investigate the toxicity mechanism of ZZHPD. The PCA and OPLS-DA-based information extracting method was used to represent the ZZHPD-induced metabolic disorders. Moreover, the underlying relationship between pathological fluctuations and metabolic disorders was elucidated according to the identified biomarkers.

## 4. Materials and Methods

### 4.1. Reagents and Chemicals

The assay kits for alanine aminotransferase (ALT), aspartate aminotransferase (AST), alkaline phosphatase (ALP), albumin (ALB), serum creatinine (CREA), urea nitrogen (BUN), total bilirubin (TBIL), total protein (TP), total cholesterol (TC), and triglyceride (TG) were purchased from Nanjing Jiancheng Bioengineering Institute (Nanjing, China). Purified water was produced by Milli-Q ultra-pure water system (Bedford, MA, USA). Formic acid (HPLC grade) was obtained from Sigma (St. Louis, MO, USA). HPLC-grade acetonitrile and methanol were purchased from Merck (Darmstadt, Germany). The standards of geniposide (purity > 98%) was purchased from Hefei Bomei Bio-technology Company (Hefei, China). Bezafibrate (purity = 99.6%) was bought from the National Institutes for Food and Drug Control (Beijing, China). Bezafibrate was used as IS for quantification.

### 4.2. Plant Material

Zhi-Zi and Zhi-Shi (collection in Jiangxi, China), and Hou-Po (collection in Sichuan, China) were purchased from Jiangsu Simcere drug store (Nanjing, China) and authenticated by Professor Ming-Jian Qin (Department of Chinese Materia Medica, China Pharmaceutical University, Nanjing, China). All of the voucher specimens were deposited in our laboratory for future reference (numbers 170631, 170615, and 170513).

### 4.3. Preparation of ZZHPD Samples

The ZZHPD sample was prepared according to the method from our laboratory [45]. The three kinds of medicinal slices were ground into powder before use. Zhi-Zi (9 g), Zhi-Shi (12 g), and Hou-Po (9 g) were percolated in water (1:10, *w*/*v*) for 0.5 h, and decocted by boiling for one hour. The extracted solution was filtered through six layers of gauze, and the residue was boiled again twice in a total volume of eight and five times (*w*/*v*) weight of these herbs, respectively. The three extracts were combined and condensed under reduced pressure. Freeze-drying was used to produce lyophilized powder, and the drug extract ratio was 4.69:1 (100%:21.3%). Appropriate amounts of the lyophilized powder were dispersed in water when administered.

### 4.4. Animals and Drug Administration

All of the protocols and care of the rats were in accordance with the guide of relevant national legislation and local guidelines. Sixty-nine male Sprague-Dawley rats (180–220 g) were purchased from Vital Laboratory Animal Technology Co. Ltd. (Beijing, China). Rats were maintained on a standard light/dark cycle under controlled temperature (25 ± 2 °C) and humidity (50 ± 10%) with a certified standard diet and water ad libitum. The animals were allowed to acclimatize for one week before oral treatment.

The rats were randomly assigned into four groups: control group (treated with 0.9% saline solution), 2.7 g/kg/day, 8.1 g/kg/day, and 27 g/kg/day ZZHPD group, with each group including six rats. The above doses were converted from the human dose into the rat dose according to body surface area, with 2.7 g/kg/day as the clinical equivalent dose (70 kg of human and 0.2 kg of rat at a conversion factor of 0.018, rat of dose = 30 × 0.018/0.2 = 2.7 g/kg). Among them, 2.7 g/kg/day is the clinical dosage, while 8.1 g/kg/day is three times the clinical dosage, and 27 g/kg/day is the maximum tolerance of the ZZHPD acute toxicity test. It has been reported that 100 mg/kg and 300 mg/kg of geniposide can produce toxic effects [16,19]. Thus, the three treatment groups of ZZHPD converting into the dose of geniposide were 30 mg/kg (2.7 g/kg/day ZZHPD group), 90 mg/kg (8.1 g/kg/day ZZHPD group), and 300 mg/kg (27 g/kg/day ZZHPD group), respectively. The four groups were given continuously gavage administration for 30 days, and blood was taken one day after the last administration. The average content of geniposide in ZZHPD was determined by UV detection at a wavelength of 238 nm and 340 mg of single-pass decoction.

For the tissue distribution, 45 rats received the dosage described above for 30 days, and 15 rats in each dose group were sacrificed at 0.5 h, 1 h, and 4 h post-dose of ZZHPD on the 30th day. Blood samples (approximately 0.5 mL) were collected from orbital venous and then centrifuged at 3000 g for 10 minutes to separate the plasma. Tissues including the liver, lung, kidney, heart, brain, and spleen were collected, washed with 0.9% (*w*/*v*) saline, and blotted dry with filter paper. Ice-cold normal saline (0.6 mL) was added to 0.2 g of tissue. Tissues were homogenized in a homogenizer. The above samples were stored at −80 °C until analysis.

### 4.5. Physical and Behavioral Changes

In the present study, the subchronic toxicity of ZZHPD was assessed used three dosages, because the ZZHPD is a long-term treatment for depression. Cage-side observation was performed to evaluate for sign of toxicity at 30 min, 1 h, and 4 h daily for 30 days. The signs of toxicity include inactivity, isolation, not grooming, abnormal gait, persistent scratching, lethargy, not eating or drinking, diarrhea, and so on. The body weights of all the animals were recorded every six days, and the administrated dosage was adjusted according to body weight. On Day 31, all of the animals were sacrificed by abdominal aortic bleeding under anesthesia using 10% chloral hydrate administered intraperitoneally at 0.35 mL/100 g.

### 4.6. Biochemical Analyses and Histopathological Examination

Ten serum biochemical parameters—AST, ALT, ALP, ALB, BUN, CREA, TBIL, TC, TP, and TG—were measured by a routine clinical laboratory method using a Clinical Chemistry Analyzer (Cobas6000, Roche, Switzerland). As the final experiment, the livers were removed immediately after sacrifice. The livers of rats were fixed in 10% neutral buffered formalin, dehydrated, embedded in paraffin, sliced to a thickness of approximately four μm, stained with hematoxylin and eosin, and examined for histological examination under the microscope (Olympus BX53, Tokyo, Japan).

### 4.7. Sample Preparation

#### 4.7.1. Untargeted Metabonomics Analysis

Prior to UPLC-Q-Exactive Orbitrap-MS analysis, plasma samples were prepared as follows: thawed at room temperature, vortexed for 30 s, then precipitated protein at the ratio of 1:3 with acetonitrile, vortexed for 60 s, centrifuged at 12,000 rpm for 10 min at 4 °C and filtered supernatant by a 0.22-µm membrane, treated as above, and then injected into the UPLC system. A total of 600 µL from each treated sample was pooled as a QC sample, and these QC samples used the same preparation method described above, respectively.

#### 4.7.2. Tissue Distribution

Fifty-µL tissue and plasma samples were prepared as follows: thawed at room temperature, vortexed for 30 s, added 20 µL of IS working solution (1 µg/mL), then, protein precipitated with 150 µL of methanol, vortexed for 60 s, and centrifuged at 12,000 rpm for 10 minutes at 4 °C. The supernatant was added to the pure water at the ratio of 1:1 and vortexed for 60 s. Finally, three µL of the mixture was injected into the HPLC-MS/MS system.

### 4.8. Instrumentation and Analytical Condition

#### 4.8.1. UPLC/MS Analysis of Metabolomics Study

Chromatographic separation was performed on a Thermo Ultimate 3000 XRS UPLC chromatographic system (Thermo Fisher Scientific, Waltham, MA, USA) with ACQUITY UPLC BEH C18 column (150 mm × 2.1 mm, 1.7 µm, Waters Corporation, Milford, CT, USA). The chromatographic conditions were described below: flow rate of 350 μL/min, column temperature of 35 °C, injection volume of three μL, sampler tray of 4 °C, mobile phase A (water + 0.1% formic acid), and mobile phase B (acetonitrile). The gradient elution programs were as follows: 0–1.5 min, 2% B; 1.5–4.0 min, 2–20% B; 4.0–9.0 min, 20–60% B; 9.0–18.0 min, 60–98% B; 18.0–19.0 min, 98% B; 19.0–19.5 min, 98–2% B; and 19.5–21.5 min, 2% B in the positive ion mode; 0–1.5 min, 2% B; 1.5–3.0 min, 2–25% B; 3.0–7.0 min, 25–45% B; 7.0–8.0 min, 45–60% B; 8.0–15.0 min, 60–98% B; 15.0–16.0 min, 98% B; 16.0–17.5 min, 98–2% B; and 17.5–19.0 min, 2% B in the negative ion mode.

MS condition was performed on a Q-Exactive Orbitrap-MS (Thermo Fisher Scientific, Waltham, MA, USA) with heated electrospray interface (HESI) in electrospray ionization (ESI). Data with a mass range of *m*/*z* 70–1050 were acquired at positive and negative modes with data-dependent MS/MS. The full scan and fragment spectra resolution were 70,000 FWHM and 17,500 FWHM, respectively. The ion source parameters were as follows: capillary temperature of 350 °C, spray voltage of 3000 V, sheath gas flow rate of 35 Arb, auxiliary gas flow rate of 15 Arb, automatic gain control (AGC) target of 1 × 10^6^ ions, and maximum ion injection time (IT) of 100 ms. Data were processed using Xcalibur™ version 3.1. (Thermo Fisher Scientific, Waltham, MA, USA). For the QC samples, pooled QC samples of data were acquired to supervise the real samples’ variability in the overall analytic run.

#### 4.8.2. HPLC-MS/MS Analysis of Tissue Distribution Study

Rat tissue samples were analyzed using an HPLC-MS/MS system and a Shimadzu LC-30 AD HPLC system (Shimadzu Corp., Tokyo, Japan). Chromatographic separation was performed on a chromatographic column Ultimate^®®^XB-C18 (50 mm × 2.1 mm, three µm, Welch, Shanghai, China), and the column temperature was maintained at 35 °C. The flow rate was set at 0.35 mL/min. The mobile phase consisted of A (water + 0.1% formic acid) and B (methanol) using a gradient elution: 0–1.30 min, 35% B; 1.30–1.40 min, 35–95% B; 1.40–2.92 min, 95% B; 2.92–3.00 min, 95–35% B; and 3.00–3.50 min, 35% B. The autosampler temperature was maintained at eight °C. The injection volume was three μL.

MS condition was performed on an AB Triple Quad 5500 type three quadrupole tandem MS with an electrospray ionization (ESI) source and Analyst 1.6.3 software (Foster City, CA, USA). The ion source parameters were as follows: spray voltage of −4500 V, source temperature of 500 °C, ion source gas one (N2) of 55 psi, ion source gas two (N2) of 50 psi, curtain gas of 20 psi. Multiple reaction monitoring (MRM) was selected for scanning mode. The following precursor and product ion for MRM were *m*/*z* 433.1 → ·225.1 (geniposide) and *m*/*z* 360.1 → ·274.1 (bezafibrate, IS). The declustering potential (DP) was set at −88 V and −82 V for geniposide and bezafibrate, respectively. The collision energy (CE) was set at −20 V and −22 V for geniposide and bezafibrate, respectively.

### 4.9. Data Processing and Biomarker Identification

Raw LC-MS data generated from the UPLC/MS analysis were processed by Xcalibur3.0 (Thermo Fisher Scientific) software. Then, these were converted to zip files, uploaded to the XCMS online platform (http://metlin.scripps.edu/xcms/) for data processing, including the nonlinear alignment of the data in the time domain and extraction of the peak intensities, the t_R_-*m*/*z* values of the control group were compared to the three ZZHPD-dosed groups by chemometrics methods. The generated data matrix was imported to SIMCA-P software (version 14.0, UmetricsAB, Umea, Sweden) and used for PCA and OPLS-DA to compare the three ZZHPD-dosed groups with the control group. The potential biomarkers differentiating the control group from the different dosage groups were selected according to VIP values. Student’s t-tests were performed for statistical analysis using SPSS 18.0 statistical software (SPSS Inc., Chicago, IL, USA) to evaluate the significant differences in potential biomarkers, with a statistical significance of *p*-value < 0.05. Variables with VIP > 1 and *p* < 0.05 were considered to be potential biomarkers. The HMDB (http://www.hmdb.ca/), METLIN (https://metlin.scripps.edu), and KEGG (http://www.genome.jp/kegg/) databases were searched for identification of the potential biomarkers.

## 5. Conclusions

In this study, it was preliminarily confirmed that ZZHPD may lead to liver damage at dosages of 8.1 g/kg/day and 27 g/kg/day after a 30-day oral administration, as indicated by the serum biochemical indices and the histopathological examination. The metabolomics analyses successfully identified 20 potential endogenous metabolites in the plasma samples that were related to the lipid metabolism and amino acid metabolism. Meanwhile, the result of the geniposide tissue distribution research indicated that the accumulation of geniposide in the liver tissue might be one of the factors that led to hepatotoxicity. Certainly, the main limitation of this work was the lack of quantitative and dynamic analyses of the metabolites in the ZZHPD-treated rats, which should be carried out in a further experimental validation.

## Figures and Tables

**Figure 1 molecules-24-00511-f001:**
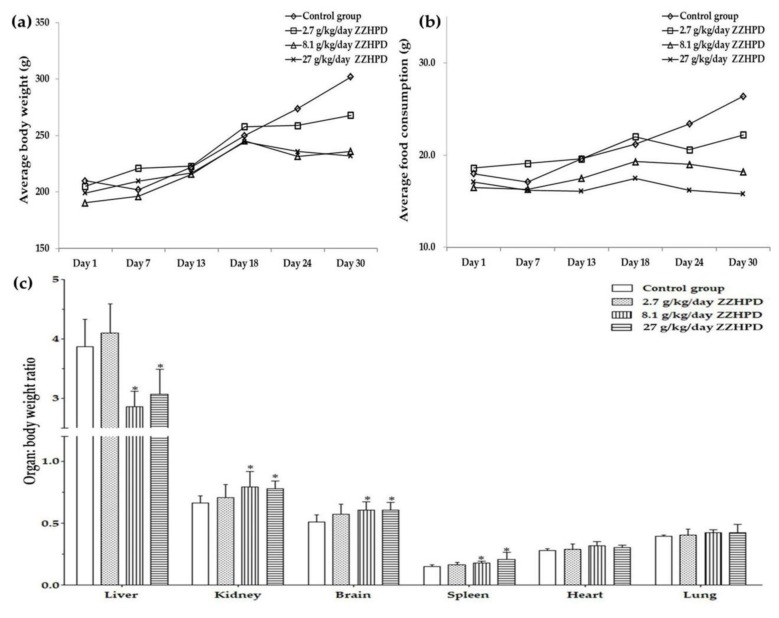
(**a**) Average body weight, (**b**) average food consumption, and (**c**) relative organ weight to body weight ratio of the liver, kidney, brain, spleen, heart, and lungs across the three dosages of the Zhi-Zi-Hou-Po decoction (ZZHPD) and the control group, where * *p* < 0.05 was considered significant compared to the control group (*n* = 6).

**Figure 2 molecules-24-00511-f002:**
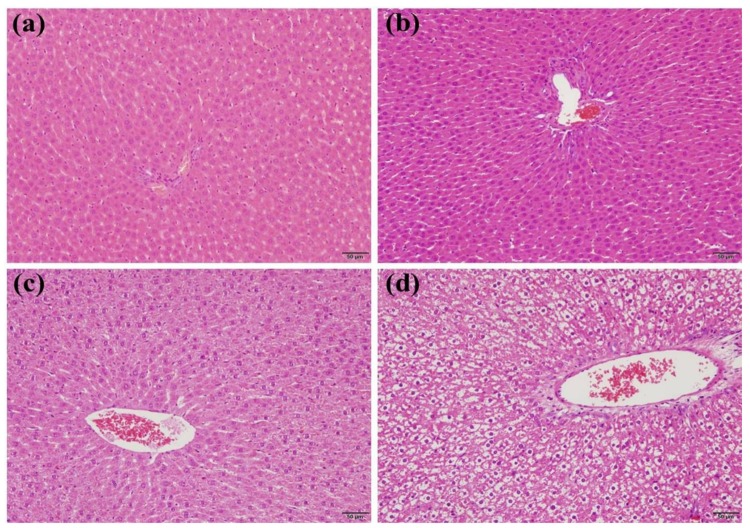
Histopathological examination of rat liver sections: (**a**) control group; (**b**) 2.7 g/kg/day ZZHPD group; (**c**) 8.1g/kg/day ZZHPD group; and (**d**) 27 g/kg/day group.

**Figure 3 molecules-24-00511-f003:**
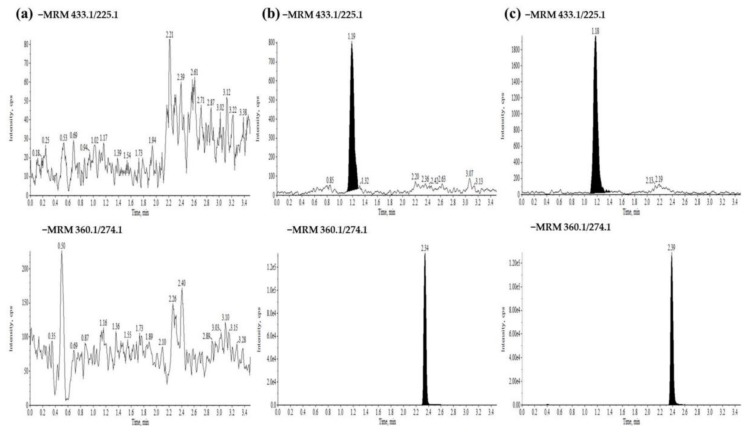
Representative multiple reaction monitoring chromatograms of geniposide and internal standards (IS) (**a**) blank tissue sample; (**b**) blank tissue sample spiked with the analytes at LLOQ concentrations and IS; and (**c**) a liver sample at four hours after oral administration of 2.7g/kg/day ZZHPD to rats.

**Figure 4 molecules-24-00511-f004:**
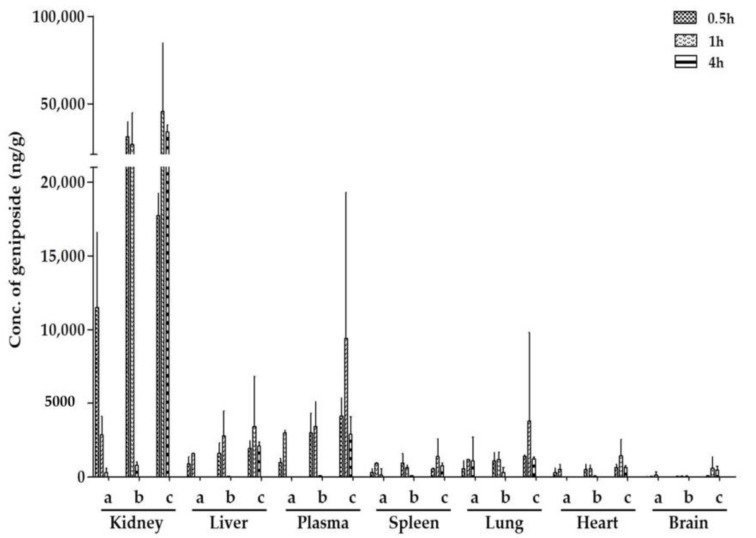
The effect of the three doses of the ZZHPD on the tissue distribution of geniposide at 0.5 hours, one hour, and four hours after treatment. (**a**) 2.7 g/kg/day ZZHPD group; (**b**) 8.1 g/kg/day ZZHPD group; and (**c**) 27 g/kg/day ZZHPD group.

**Figure 5 molecules-24-00511-f005:**
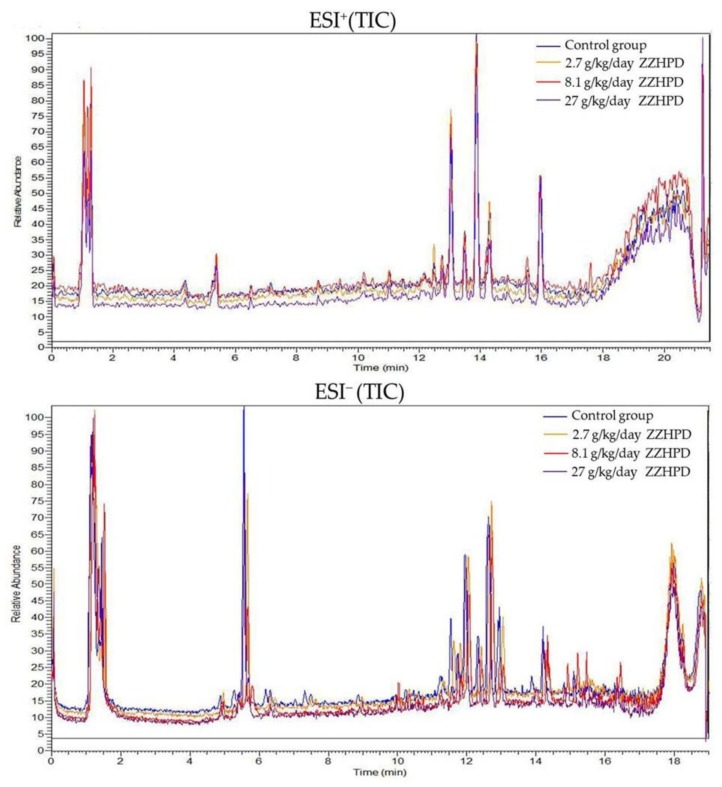
Typical total ion chromatograms (TIC) of plasma samples obtained from UPLC-MS analysis.

**Figure 6 molecules-24-00511-f006:**
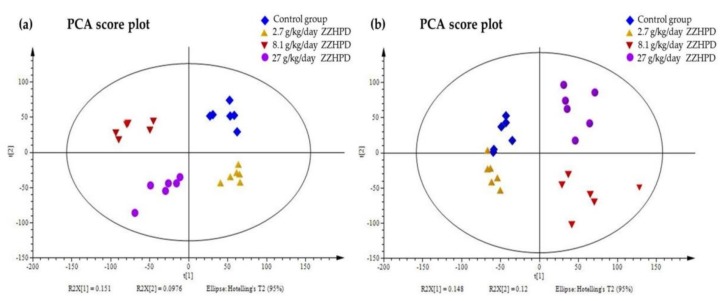
The principal components analysis (PCA) score plot of the rat plasma metabolites. (**a**) In the positive ion mode; (**b**) In the negative ion mode.

**Figure 7 molecules-24-00511-f007:**
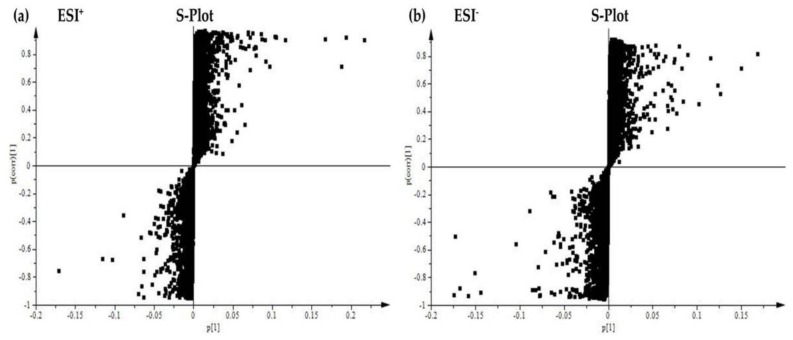
The partial least-squared discriminant analysis (PLS-DA) S-plot of the rat plasma metabolites. (**a**) In the positive ion mode; (**b**) In the negative ion mode.

**Figure 8 molecules-24-00511-f008:**
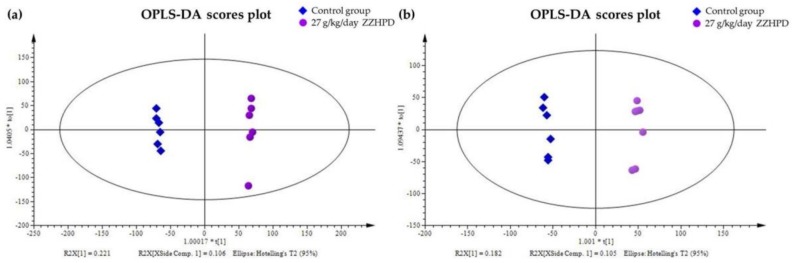
The orthogonal partial least-squared discriminant analysis (OPLS-DA) scores plot of the rat plasma metabolites. (**a**) In the positive ion mode; (**b**) In the negative ion mode.

**Figure 9 molecules-24-00511-f009:**
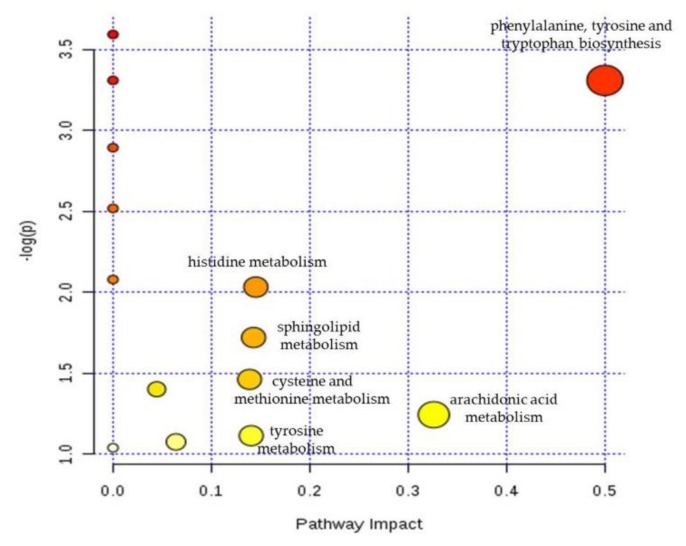
The potential metabolic pathways according to the identified biomarkers using MetaboAnalyst 4.0 analysis.

**Figure 10 molecules-24-00511-f010:**
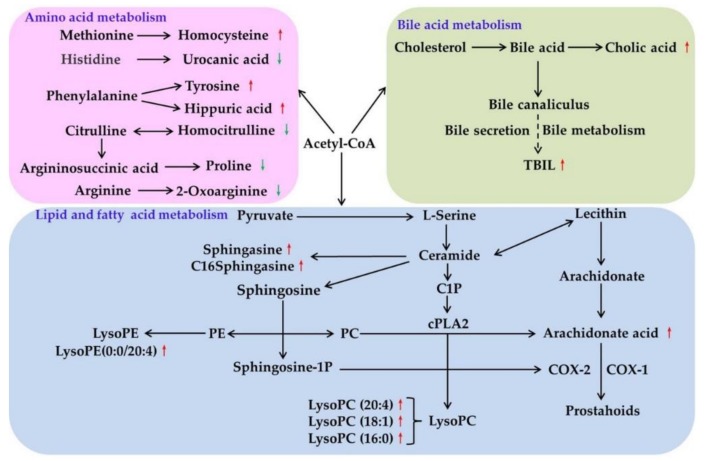
Schematic diagram of the perturbed metabolites and biochemical index corresponding metabolic pathways related to the ZZHPD treatment. The notations were as follows: elevation (up and red thin arrows) and reduction (down and green thin arrows) of the levels of metabolites and biochemical index observed in rat plasma and serum were indicated.

**Table 1 molecules-24-00511-t001:** Serum biochemical parameters that were orally administered different dosages of ZZHPD and saline solution for 30 days (mean ± SD, *n* = six/group).

Group	Biochemical Parameters									
	ALT (U/L)	AST (U/L)	ALP (U/L)	ALB (g/L)	CREA (µmoL/L)	BUN (mmoL/L)	TBIL (µmoL/L)	TP (g/L)	TC (mmoL/L)	TG (mmoL/L)
Control	217.41 ± 51.80	172.22 ± 65.45	182.87 ± 21.60	29.39 ± 3.31	44.08 ± 5.24	7.27 ± 1.58	0.52 ± 0.28	48.57 ± 2.09	3.34 ± 0.72	2.00 ± 0.80
2.7 g/kg/day (ZZHPD)	214.07 ± 30.43	168.15 ± 37.89	173.28 ± 14.34	28.94 ± 1.83	41.50 ± 2.99	6.42 ± 1.22	0.45 ± 0.10	48.82 ± 1.67	3.27 ± 0.30	1.36 ± 0.42
8.1 g/kg/day (ZZHPD)	501.48 ± 62.22 **	703.33 ± 67.15 **	73.74 ± 17.85 **	27.17 ± 1.36	27.51 ± 4.06 **	11.36 ± 2.23 *	3.98 ± 0.93 **	42.37 ± 1.56 **	6.86 ± 1.28 **	3.01 ± 0.76
27 g/kg/day (ZZHPD)	838.89 ± 167.82 **	985.93 ± 182.80 **	75.40 ± 17.34 **	25.68 ± 2.12 *	26.06 ± 3.60 **	12.65 ± 0.93 **	4.91 ± 0.69 **	41.81 ± 1.42 **	8.24 ± 0.57 **	3.43 ± 0.37 **

Compared to the control group: * *p* < 0.05, ** *p* < 0.01 (one-way ANOVA). ALT: alanine aminotransferase, AST: aspartate aminotransferase, ALB: albumin, CREA: serum creatinine, BUN: urea nitrogen, TBIL: total bilirubin, TP: total protein, TC: total cholesterol, TG: triglyceride.

**Table 2 molecules-24-00511-t002:** A list of significant biomarkers and their association with the hepatoxicity of ZZHPD.

No.	Rt/min	*m*/*z*	ESI Mode	Biomarker Identification	Formula	VIP	2.7 g/kg/day	8.1 g/kg/day	27 g/kg/day	Dysfunction Association
1	1.18	144.1015	+	Vinylacetylglycine	C_6_H_9_NO_3_	2.09	-	↑↑↑	↑↑↑	fatty acid metabolism
2	11.95	588.3292	-	LysoPC (20:4)	C_28_H_50_NO_7_P	2.06	-	↑	↑↑↑	lipid metabolism
3	14.81	327.2321	-	Docosahexaenoic acid	C_22_H_32_O_2_	2.05	-	↑↑	↑↑	lipid metabolism
4	7.36	201.0216	-	Bergaptol	C_11_H_6_O_4_	2.04	↓	↓↓	↓↓	energy metabolism
5	13.01	544.3386	+	LysoPC (18:1)	C_26_H_52_NO_7_P	2.02	-	↑↑	↑↑↑	lipid metabolism
6	10.14	274.2733	+	C16 Sphinganine	C_16_H_35_NO_2_	1.99	-	↑↑↑	↑↑↑	lipid metabolism
7	1.14	138.0545	+	Urocanic acid	C_6_H_6_N_2_O_2_	1.96	-	↓↓↓	↓↓↓	amino acid metabolism
8	17.26	305.2457	+	Arachidonic acid	C_20_H_32_O_2_	1.95	-	↑↑	↑↑↑	lipid metabolism
9	15.36	279.2321	-	Octadecadienoate	C_18_H_32_O_2_	1.90	-	↑↑	↑↑	lipid metabolism
10	1.15	116.0706	+	Proline	C_5_H_9_NO_2_	1.89	-	↓↓	↓↓	amino acid metabolism
11	7.14	174.091	+	2-Oxoarginine	C_6_H_11_N_3_O_3_	1.87	↑	↓↓	↓↓	amino acid metabolism
12	7.52	186.0549	-	Indoleacrylic acid	C_11_H_9_NO_2_	1.87	-	↓	↓↓	amino acid metabolism
13	11.12	302.3045	+	Sphinganine	C_18_H_39_NO_2_	1.86	↑↑	↑↑	↑↑↑	lipid metabolism
14	11.91	500.277	-	LysoPE (0:0/20:4)	C_25_H_44_NO_7_P	1.84	-	-	↑	fatty acid metabolism
15	8.69	190.0861	+	Homocitrulline	C_7_H_15_N_3_O_3_	1.70	-	↓↓	↓↓	amino acid metabolism
16	1.29	136.0755	+	Homocysteine	C_4_H_9_NO_2_S	1.68	-	↑	↑↑	amino acid metabolism
17	1.29	182.0807	+	Tyrosine	C_9_H_11_NO_3_	1.66	-	↑	↑↑	amino acid metabolism
18	12.6	540.3295	-	LysoPC(16:0)	C_24_H_50_NO_7_P	1.42	-	↑	↑↑	lipid metabolism
19	9.94	407.2794	-	Cholic acid	C_24_H_40_O_5_	1.33	-	-	↑	bile acid metabolism
20	5.44	178.0498	-	Hippuric acid	C_9_H_9_NO_3_	1.30	-	↑	↑	amino acid metabolism

↑: the compound is up-regulated. ↓: the compound is down-regulated; ↑ or ↓ indicated significant difference at *p* < 0.05, compared with the control group. ↑↑ or ↓↓ indicated significant difference at *p* < 0.01, compared with the control group. ↑↑↑ or ↓↓↓ indicated significant difference at *p* < 0.001, compared with the control group.

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
