# Peer review of "Evaluation of the Hepatotoxicity of the Zhi-Zi-Hou-Po Decoction by Combining UPLC-Q-Exactive-MS-Based Metabolomics and HPLC-MS/MS-Based Geniposide Tissue Distribution"

_molecules, 2019, doi:10.3390/molecules24030511_

Round 1
Reviewer 1 Report
Authors evaluated ZZHPD-induced hepatotoxicity in rats. In this study, authors clarified the change of serum biochemical parameters, histopathological changes in liver section, changes of various metabolites related to amino acid, lipid, and bile acid metabolism. The paper included valuable information to understand ZZHPD-induced hepatotoxicity in humans. However, there are some concerns as bellow.
(1) I could understand that geniposide is main bioactive component in ZZHPD. However, the concentration in the liver is similar to that in plasma, although authors mentioned “geniposide had accumulated in the liver that may be the basis for the production of liver toxicity (page 14, line287). I wonder whether geniposide alters serum biochemical parameters, histopathological images in liver section and various metabolites. Authors should directly evaluate relationship between geniposide and these findings. Other components may contribute to the hepatotoxicity. Rather, geniposide may contribute to kidney toxicity because high concentrations were exposed in kidney (Fig.4).
(2) ALT, AST, ALT, and TBIL are the main indicators of liver as authors described (page 3, line 93). However, ALP values decreased dose-dependent manner although ALT, AST, TBIL increased (Table 1). Please explain why only ALP decreased.
(3) Authors described that “twenty compounds were identified among the four groups and control group (P < 0.05). Here, twenty compounds were identified among the four groups as hepatotoxicity of ZZHPD biomarkers (page 10, line 235)”. In these twenty compounds, were the components in ZZHPD or molecules (metabolites) derived from ZZHPD included? I think that it difficult to distinguish between ZZEPD components and endogenous compounds related to hepatotoxicity.
Reviewer 2 Report
The manuscript's English language needs improvement. There are a lot of grammatically incorrect sentences, with many syntax and diction errors. This makes the manuscript difficult to understand. Please have the manuscript edited and revised by someone with scientific English language fluency and for scientific accuracy.
Replace “normal”, “medium” and “high” with actual numerical dosages.
The authors provide a lot of speculation on potential roles in hepatotoxicity of the 20 compounds that were found at higher levels in plasma of geniposide treated rats. Cox2 RNA/protein expression levels should be determined to support the speculations of the mechanistic actions of these compounds contributing to hepatotoxicity, as well as expression of other signaling pathways implicated by the authors. In addition hepatic levels of genipin should be measured.
Author Response
Reviewer 2#:
Point 1: The manuscript's English language needs improvement. There are a lot of grammatically incorrect sentences, with many syntax and diction errors. This makes the manuscript difficult to understand. Please have the manuscript edited and revised by someone with scientific English language fluency and for scientific accuracy.
Response 1: This manuscript’s English language has been improvement and the revised part of the manuscript has been marked red.
Point 2: Replace “normal”, “medium” and “high” with actual numerical dosages.
Response 2: The “normal”, “medium” and “high” of the manuscript have been altered to actual numerical dosages.
Point 3: The authors provide a lot of speculation on potential roles in hepatotoxicity of the 20 compounds that were found at higher levels in plasma of geniposide treated rats. Cox2 RNA/protein expression levels should be determined to support the speculations of the mechanistic actions of these compounds contributing to hepatotoxicity, as well as expression of other signaling pathways implicated by the authors. In addition hepatic levels of genipin should be measured.
Response 3: In this study, a untargeted metabolomics approach based on high resolution liquid chromatography-mass spectrometry in conjunction with multivariate statistical data analyses was employed to determine global alterations in the metabolic profiles. Based on the discovered toxic biomarkers, we considered using the method of targeted metabolomics to quantitatively analyze the changes of small molecular biomarkers in the process of liver injury in pathway for next step, so as to predict the occurrence of liver injury caused by ZZHPD. The determination of COX-2 RNA/protein expression levels mentioned by the reviewers are good suggestions for our metabolome test validated. It can be combined with endogenous small molecular biomarkers for some metabolic pathway to elucidate the occurrence of ZZHPD liver injury. Meanwhile, the conclusion of the manuscript also acknowledges the shortcomings of the article (page 19, line 526). Although this study did not verify all of the metabolic pathways, some metabolic pathways can also be confirmed from the literature [1.2], such as “Dose-related liver injury of Geniposide associated with the alteration in bile acid synthesis and transportation”. The increase of biochemical index TBIL (Table 1) and cholic acid (Table 2) in this study can also support the view that ZZHPD induced liver injury involving bile acid metabolism.
Because the manuscript was only modified for 10 days, we didn’t able to determine the concentration of genipin in the liver and kidney. Although this study did not measure hepatic levels of genipin, we searched some relevant literature to support this result, such as “The study of geniposide on toxicokinetics in rats [3]”. It was said that geniposide was converted to genipin after oral administration [4], and geniposide and genipin significantly were increased the concentration of liver tissue at different doses. In these measured tissues of geniposide and genipin in the order of kidney> plasma> liver > spleen > lung> heart> brain and liver > plasma > kidney > spleen > heart > lung > brain, respectively. This result was in agreement with the discussion of manuscript that “the geniposide and genipin accumulated in the liver lead to toxicity reaction” (page 14, line 300) and “The result indicated that geniposide was distributed rapidly and widely in these measured tissues in the order of kidney > plasma > liver > lung > spleen≈ heart > brain of the three dosages (page 7, line 167).” At the same time, we had modified the discussion section of the manuscript (page 14, line 295) in order to better understand why the high concentration of geniposide in the kidney was not found to be nephrotoxic.
References
[1] Tian, J.Z.; Zhu, J.J.; Yi, Y.; Li, C.Y.; Zhang, Y.S.; Zhao, Y.; Pan, C.; Xiang, S.X.; Li, X.L.; Li, G.Q.; Newman, JW.; Feng, X.Y.; Liu, J.; Han, J.Y.; Wang, L.M.; Gao, Y.; La Frano, MR.; Liang A.H. Dose-related liver injury of Geniposide associated with the alteration in bile acid synthesis and transportation. Sci Rep. 2017, 7, 8938. https://doi.org/10.1038/s41598-017-09131-2.
[2] Dong, L.C.; Fan, Y.X.; Yu, Q.; Ma, J.; Dong, X.; Li, P.; Li, H.J.Synergistic effects of rhubarb-gardenia herb pair in cholestatic rats at pharmacodynamic and pharmacokinetic levels. J Ethnopharmacol.2015, 175, 67-74. https:// doi.org/10.1016/j.jep.2015.09.012.
[3] She, D. The study of geniposide on toxicokinetics in rats, Guangzhou University of Chinese Medicine, Guangzhou, 2015; pp.36.
[4] Shi, F.G.; Pan, H.; Li, Y.; Huang, L.Y.; Wu, Q.; Lu, Y.F. A sensitive LC–MS/MS method for simultaneous quantification of geniposide and its active metabolite genipin in rat plasma and its application to a pharmacokinetic study. Biomed Chromatogr. 2018, 32. https://doi.org/10.1002/bmc.4126.

Round 2
Reviewer 1 Report
I think that authors have revised the manuscript according to reviewer comments.
Author Response
Dear Editors and Reviewers:
Thank you for your letter and for the reviewers’ comments concerning our manuscript entitled “Evaluation of the hepatotoxicity of the Zhi-Zi-Hou-Po decoction by combining UPLC-Q-Exactive-MS-based metabolomics and HPLC-MS/MS-based geniposide tissue distribution” (ID: molecules-419274).
In view of the shortcomings of English language and style in this manuscript, we had adopt the MDPI English editing to further improve the language of the article (English editing ID: english-7655). We tried our best to improve the manuscript and made some changes in the manuscript that marked red. We hope our efforts can improve the quality of the manuscript.
We appreciate for Editors and Reviews’s warm work earnestly, and hope that the correction will meet with approval.
Once again, thank you very much for your comments and suggestions.
Reviewer 2 Report
Though there is some improvement in English, still extensive improvement is required to make sure the manuscript can be understood by the audience of the journal. Grammatical mistakes and/or syntax mistakes in almost every sentence. Therefor this comment has been far from sufficiently addressed.
Author Response

(The authors gave the same response as above.)
